# Flow Boiling of Liquid n-Heptane in Microtube with Various Fuel Flow Rate: Experimental and Numerical Study

**DOI:** 10.3390/mi14091760

**Published:** 2023-09-12

**Authors:** Muhammad Tahir Rashid, Naseem Ahmad, Raees Fida Swati, Muhammad Bilal Khan

**Affiliations:** 1School of Aerospace Engineering, Beijing Institute of Technology, Beijing 100081, China; 2Department of Mechanical Engineering, Institute of Space Technology, Islamabad 44000, Pakistan; 3Department of Aeronautics and Astronautics, Institute of Space Technology, Islamabad 44000, Pakistan; 4Faculty of Mechanical Engineering, Ghulam Ishaq Khan Institute of Engineering Sciences and Technology, Topi, Swabi 23640, Pakistan

**Keywords:** volume of fluid, microtube, evaporating meniscus, flow boiling, pressure drop fluctuation, heat transfer

## Abstract

The evaporation of liquid hydrocarbon n-heptane is discussed in detail with experimentation and numerical techniques. A maximum wall temperature of 1050 K was reported during an experimental process with a two-phase flow that was stable and had a prominent meniscus at a small fuel flow rate (FFR) ≤ 10 µL/min. At medium to high FFR (30–70 µL/min), the flow field was unstable, with nucleating bubbles and liquid droplets inside the microtube and the maximum temperature recorded was 850 K for 70 µL/min. For the numerical model, the temperature of the wall was used as a boundary condition. Using the numerical model, the evaporative flux at the meniscus, pressure drop, pressure oscillation, and heat transfer coefficient (HTC) were investigated. A single peak in HTC was obtained at a low fuel flow rate, while multiple peaks were obtained for high FFR. At low FFR, the pressure peak was observed to be 102.4 KPa, whereas at high FFR, the pressure peak increased to 105.5 KPa. This shows a 2% increase in pressure peak with an increase in FFR. Similarly, when the FFR increased from 5 µL/min to 70 µL/min, the pressure drop increased from 500 Pa to 2800 Pa. The high amplitude of pressure drops and a high peak of HTC were found, which depend on the mass flow rate. The coefficient of variation for pressure drop depends mainly on the fuel flow rate.

## 1. Introduction

Numerous research efforts are directed towards the manufacturing and downsizing of micro-scaled devices, including micro gas turbines [1,2,3], micro swing engines [4,5,6,7], micro Swiss-roll combustors [8,9], and micro photovoltaic devices [10,11], and have attracted considerable research for further development. The typical power source for such systems is batteries, but due to weight restrictions on micro devices and the low energy density of batteries, researchers are seeking alternate methods of producing electricity on a small scale while utilizing the micro combustor to benefit from the high energy density (40 KJ/g) of hydrocarbon fuel. This energy density is approximately 100 times greater than that of the majority of sophisticated batteries [12,13,14,15]. However, it is noted that flow boiling and flame stability are critical concerns in the development of micro combustors [16,17].

Several experimental studies [18,19,20] have focused on microchannel boiling heat transfer and flow pattern. Leong et al. [18] gave a critical review of the boiling model and bubble dynamics in the microchannel. Parjapati et al. [19,21] reported the instabilities in micro channels and also described their promising features, whereas Cebi et al. [20] worked on flow boiling in mini and microchannels for enhanced geometries. Both are time-consuming and expensive due to their design and operating techniques. Furthermore, the properties inside the microtube cannot be measured correctly due to its small size. Due to the above reasons, numerical simulation is a widely used technique for such purposes.

A few theoretical models for microchannel flow patterns were also proposed with empirical correlation and simplifications in the past. Thome [22,23] proposed a theoretical model that accounts for surface tension, gravity, the near-wall effect, wall conduction, axial shear stress, and the unstable term to calculate the heat transfer caused by the presence of a liquid layer in the mini and microchannel. Zhao and Liao’s [24] research was based on a theoretical model for vapor flowing in a vertical mini-triangular tube to condense into a film. Due to the thin liquid covering on the sidewall of the triangular channel and the surface tension effect, they revealed that the HTC at the cross-section is higher than the circular channel with similar characteristics. Wang and Rose [25,26] developed a mathematical model for a non-circular microchannel that took into account gravity, interfacial shear stress, and surface tension. They elucidated that the surface tension effect is what causes the increase in heat transfer.

Several numerical simulations of flow boiling have been also reported in recent years. Numerical simulation in square microchannels was conducted by Ferrari et al. [27], who demonstrated that a bubble moves more quickly in a square microchannel than in a tube. Additionally, they enhanced flow mixing, which affected the efficacy of the heat transfer. The annular flow boiling in microchannels was studied by Guo et al. [28] using a numerical model and it was revealed that a modest input perturbation has very little impact on boiling heat transfer. Moreover, the authors showed that uncertainties at the interface caused by phase change have a substantial influence on the rate of heat transfer via interfacial wave advection. A three-dimensional numerical simulation on a 0.6 mm microtube was conducted by Tiwari et al. [29]. They discovered that the relationship between the thermal conductivity of the microtube wall and the fluid is crucial to the behavior of bubbles during their formation and that the thickness and material of the microtube wall change during bubble confinement. The influence of hydrogen and carbon dioxide on biogas was studied by Ahmed et al. [30,31]. They revealed that the utilization of biogas with diesel is attractive to cut down discharges and enhance the performance of the engine.

Liquid hydrocarbon flow boiling has drawn a lot of interest because of its high energy density, low storage requirements, ease of transportation, and small space requirements for microenergy systems. Muller et al. [32,33] carried out an experimental study in an annulus with heptane fluid to evaluate the sub-cooled boiling and HTC in a convective regime. They noticed several convective and nucleate boiling heat transfer regimes. The heat transfer coefficients for nucleate boiling were discovered to be independent of the mass velocity. The boiling hysteresis phenomenon was only observed at small mass velocities, and the flow direction’s influence was not detected. Hapke et al. [34] employed a thermographic approach to examine how n-heptane flows and boils in a microchannel. They observed temperature profiles over the width of the channel and noticed different boiling regimes. Moreover, they concluded that both heat and mass flux disturb pressure drop in the microchannel. However, they argued that numerical simulation should be implemented to solve these types of problems. A numerical simulation of the influence of injection time was conducted [35,36] and it was found that near the contact line, large diffusion fluxes exist due to the intensity of thin film evaporation, and local interfacial temperature and evaporative flux are slightly increased. Very little literature is available for the numerical investigation of liquid n-heptane in a microtube.

The main aim of the current research is to develop a numerical model for understanding flow boiling in the microtube, the interfacial evaporation rate, and the oscillation of liquid pressure. Non-uniform temperature supplied to the inner wall causes the flow to boil. Additionally, the non-uniform temperature is measured experimentally with an infrared camera.

## 2. Experimental Setup

The experimental setup contains a straight microtube, a delivery system for fuel, a temperature measurement system, and an observation system, and is shown in Figure 1. The microtube is mounted horizontally on a syringe during experimentation. In this experiment, quartz (QT) with an outer diameter of 0.5 mm and an interior diameter of 0.2 mm is employed. Liquid fuel is injected using a syringe pump (LSP01-1A) and a plastic syringe with an inner diameter of 10 mm. The steel tube is first connected to the syringe and then the QT tube is glued with a steel tube. There is a small diffusion flame anchored at the tube’s exit. Every experiment was carried out in a single environment and at room temperature.

The syringe pump precisely regulates the fuel flow rate with an inaccuracy of less than 1.0%. The camera captures the tiny flame at 128 fps. An infrared radiation (IR) camera (FLIR, A40) is used to record the surface temperature of the microtube. The IR camera has a 150 mm macro lens and a 64 mm by 48 mm field of vision. The spectral range that was measured is 7.5 to 13 μm. The temperature variations on the tube surface are shown in the IR images. The calibration of the IR camera is discussed in Tahir et al. [17].

## 3. Observation

N-heptane is employed as a liquid fuel in the experiment, and the flow rate varies from 5 μL/min to 70 μL/min. Figure 2a depicts that the flame shape is spherical with a flame length of ~1 mm at Qf ≤ 10 μL/min. Figure 2a also depicts the equivalent liquid–vapor interface with segregated liquid and vapor phases to demonstrate the phenomenon of laminar evaporation. The two phenomena are captured by an ordinary camera at 128 fps. The instantaneous times of 0.032 s and 0.007 s show the capture time of the flame oscillation and flow pattern oscillation, respectively. A linear trend can be seen in the succeeding flame length and height depicted in Figure 2b along with the interface distance. This suggests that laminar evaporation in the microtube causes a continuous flame to form at low fuel flow rates. Due to a low fuel flow rate, the liquid n-heptane is entirely absorbed in the liquid–vapor interface. The interface distance was approximately 2.30–2.35 mm and 2.14–2.25 mm, respectively, at FFR of 5 μL/min and 10 μL/min. Nucleate bubbles and liquid droplets were not observed in the subcooled liquid phase or vapor phase, respectively.

The flame is stretched out and no longer resembles a ball when Qf is less than 30 μL/min. Because of the incomplete combustion of the hydrocarbons, the flame is more yellow and is referred to as unstable. The blue light-containing lower portion of the flame (Figure 3a) extends at 0.302 s, which may be due to the burning of liquid droplets. Figure 3a illustrates the flame at Qf 70 μL/min, which has an extensive reaction zone and a bright yellow color from the burning of slugs or liquid droplets. Additionally, the flame is referred to as an explosive flame even though it has no particular configuration and burns violently.

A high-speed camera at 8000 fps is used to record the evaporation phenomena with FFR > 30 µL/min. The two-phase flow phenomena with FFR > 30 µL/min are depicted in Figure 3. Convective boiling predominates with an FFR of 30 µL/min, and we did not observe the commencement of nucleation boiling (ONB), but when the FFR is increased to 70 µL/min, nucleate boiling is observed. Figure 3b depicts the progression of the nucleate bubble’s rupture cycle from 18.75 ms to 19.37 ms. Figure 3 shows the liquid droplet at FFR > 30 µL/min. As shown in Figure 2a, a stable flame forms at the tube exit when the FFR is small because the fuel is completely evaporated at the interface. However, as the FFR increases, more liquid fuel must be vaporized, but the liquid–vapor interface becomes too narrow and small, making it impossible to vaporize excess liquid. As shown in Figure 3, the extra liquid fuel may vaporize as a continuous liquid film along the tube’s interior wall or as a liquid droplet in the vapor domain.

While the evaporation changes from laminar to nucleate boiling with an increase in FFR, the flame shape changes from a steady flame to an explosive flame. The temperature distribution along the micro tube’s exterior wall is depicted in Figure 4. At a low fuel flow rate, the maximum temperature gradient is 350 K/m. As the FFR increases to medium flow conditions, the temperature gradients increase up to 14%. A further increase in flow conditions to a higher FFR results in a 65% increase compared to the gradient of low FFR. The percentage increase in the temperature gradient from low to medium FFR is small, i.e., 14%, and from medium to high FFR the percentage increase is 65%. The maximum temperature of the wall drops as the FFR rises, and this is due to two factors. The first explanation is that a high FFR results in greater heat absorption for gasification, which lowers the temperature of the wall. The second reason is that the presence of liquid slugs and droplets in the vapor domain lowers the wall temperature by requiring more heat for gasification.

The aforementioned observation leads to the hypothesis that FFR has impacted flame, which, in turn, has impacted wall temperature, which then has impacted interface movements. In order to study the impact of fuel flow rate and wall temperature on two-phase flow, pressure drop, pressure oscillation, and heat transfer, numerical simulation is conducted in Ansys 16.

## 4. Mathematical Modeling

The following assumptions are made in the current research:In the microtube, the gravity effect is dominated by the surface tension force.The physical properties like surface tension, latent heat, and density are constant for each phase.The liquid n-heptane boiling point is assumed to be 373.15 K.

### 4.1. VOF Model

The Volume of Fluid (VOF) method in the commercial software Fluent 16.0 with a user-defined function (UDF) was used to describe two-phase flow boiling. A fixed grid method called the VOF model was created for two or more immiscible fluids. The VOF model resolves a single set of fluid-specific momentum equations. The VOF method records the volume fraction of each phase in the computing cell, and the total volume fraction of all phases equals unity.
(1)$$\alpha_{l}+\alpha_{v}=1$$

The equation for the vapor and liquid phase is given as:(2)$$\frac{{\partial \alpha }_{l}}{\partial t}+\nabla .\left(\overset{⃑}{v}\alpha_{l}\right)=-\frac{S}{\rho_{l}}$$
(3)$$\frac{{\partial \alpha }_{v}}{\partial t}+\nabla .\left(\overset{⃑}{v}\alpha_{v}\right)=-\frac{S}{\rho_{v}}$$

The continuity, energy, and momentum equations are:(4)$$\frac{\partial (\rho v)}{\partial t}+\nabla .\left(\rho \overset{⃑}{v}\overset{⃑}{v}\right)=-\nabla p+\nabla .\left[\mu \left(\nabla \overset{⃑}{v}+\nabla {\overset{⃑}{v}}^{T}\right)-\frac{2}{3}\mu \nabla .\overset{⃑}{v}I\right]+\rho \bar{g}+F_{vol\;}$$
(5)$$\frac{\partial (\rho E)}{\partial t}+\nabla .\left[\overset{⃑}{v}(\rho E+p)\right]=\nabla .\left(K\nabla T\right)+Q$$
where
(6)$$\rho =\alpha_{l}\rho_{l}+\alpha_{v}\rho_{v}$$
(7)$$k=\alpha_{l}k_{l}+\alpha_{v}k_{v}$$
(8)$$\mu =\alpha_{l}\mu_{l}+\alpha_{v}\mu_{v}$$
(9)$$\;\;\;\;\;\;\;\;E=\frac{\alpha_{l}\rho_{l}E_{l}+\alpha_{v}\rho_{v}E_{v}}{\alpha_{l}\rho_{l}+\alpha_{v}\rho_{v}}$$

The Continuum Surface Force (CFS) model is used for the surface tension effect, and surface tension can be described as a pressure jump over the surface. The divergence theorem can be used to express the force at the surface into a volume force, *F_vol_*. The momentum equation is represented as:(10)$$F_{vol}=\sigma_{lv}\frac{\alpha_{l}\rho_{l}k_{v}\nabla \alpha_{v}+\alpha_{v}\rho_{v}k_{l}\nabla \alpha_{l}}{0.5(\rho_{l}+\rho_{v})}\;$$
where *σ_lv_* is the surface tension at the interface. The definition of curvatures of the liquid and vapor phase is defined as:(11)$$k_{l}=\frac{∆\alpha_{l}}{\left|∆\alpha_{l}\right|}\;\;\;\;\;\;\;\;k_{v}=\frac{∆\alpha_{v}}{\left|∆\alpha_{v}\right|}$$

### 4.2. Heat and Mass Transfer Due to Evaporation at the Interface

The Lee model [37] is used for mass transfer and predicts phase transition at constant pressure. In a quasi-thermo-equilibrium condition, the key determinant of mass transfer is saturation temperature. The Lee model is defined as:(12)$$S=\frac{r_{l}\alpha_{l}\rho_{l}\left(T_{l}-T_{sat}\right)}{T_{sat}}\;\;\;\;\;T_{l}{\geq T}_{sat}\;$$
(13)$$=\frac{r_{v}\alpha_{v}\rho_{v}\left(T_{l}-T_{sat}\right)}{T_{sat}}\;\;\;\;\;T_{l}{<T}_{sat}$$

The parameters *r*_l_ and *r_v_* are equal to 100 s^−1^ in order to maintain the interface temperature in the range of 1 K [28,38,39]. Small values of these coefficients produce a significant difference between saturation and interfacial temperature, while excessively large values cause numerical divergence. The direct measurements of the heat transmission are:(14)$$Q=-h_{lh}S$$

The following equation is used to determine the interface evaporative flux [40]
(15)$${\ddot{m}}_{eva}=\left[\frac{2\;.\;\bar{\sigma }}{2-\bar{\sigma }}\right]\;.\;{\left(\frac{MB_{vap}}{2\;.\;\pi \;.\;\bar{R}}\right)}^{1/2}.\left(\frac{P_{sat,liq}}{T_{liq}^{1/2}}-\frac{P_{sat,vap}}{T_{vap}^{1/2}}\right)$$

### 4.3. Simulation Condition and Grid System

The computational domain consists of a 2D axisymmetric with both the solid and liquid regions having the same specification as the quartz tube used in the experiments, as depicted in Figure 5. The length of the tube is 5 mm as the major temperature variation is in this range. This small length also gives benefits to fewer computation resources. The surface tension force’s dominance over gravity at a small tube diameter is confirmed using the Bond number criterion [41]. The gravitational impact was therefore neglected, and the 2D calculation region is validated. Table 1 provides the heptane characteristics.

For an FFR of 5 µL/min, the grid independence analysis was carried out. In the current research work, a quadrilateral mapped mesh is employed and it is shown in Figure 6. The same mesh size is used for both the solid and liquid parts. Table 2 represent a different grid for both solid and liquid domain at FFR of 5µL/min. To calculate the relative error (*e*%) the following equation is used:(16)$$e\%=\left|\frac{X_{1}-X_{2}}{X_{1}}\right|\times 100$$
where *X*_1_ is the parameter value obtained from the finest grid; *X*_2_ is the value obtained from other grids; and *X_i_* (*i* = 1, 2, 3...) shows the variable of interest, such as the rate of evaporation at the interface, the average heat transfer coefficient, or the pressure drop. The grid independence analysis was carried out at a mass flow rate of 5 µL/min. Three different mesh types are used in the current research. Table 2 provides the mesh detail information.

Based on the above test results, mesh 2 will be employed in all simulations as adapted because the deviation from mesh 1 is minimal, and also in this mesh, the computational cost is low. The edge length along the *x*-axis is 4.5 µm, while the edge length in the *y*-axis is 4 µm. The courant number, which changes depending on the fuel flow rate, determines the time step. Table 3 provides the volumetric flow rate of fuel and the corresponding mass flow rate.

The geometric reconstruction scheme [38] is employed for a volumetric fraction. Pressure-implicit with the splitting of operators, or PISO, is used to couple pressure and velocity because it provides a more approximate relationship between the two corrections. The momentum and energy equations were discretized using an upwind approach of the first order.

### 4.4. Data Reduction

The microtube’s outer surface receives heat addition and is given by:(17)$$Q=\dot{m}c_{p}\left(T_{out}-T_{in}\right)$$

The heat flux at the microtube’s outer surface is calculated using energy balance and is stated as:(18)$$q_{applied}^{\prime \prime }=\frac{Q}{A_{s}}=\frac{\dot{m}c_{p}(T_{out}-T_{in})}{2\pi r_{o}L}$$

The definition of the local Nusselt number is:(19)$${Nu}_{l}=\frac{2\delta_{f}h_{l}}{\lambda_{f}}$$
where *λ_f_* is n-heptane’s thermal conductivity and *h_l_* is the local heat transfer coefficient. To calculate the local heat transfer coefficient as:(20)$$h_{l}=\frac{q_{z}^{\prime \prime }}{T_{w}-T_{f}}\;$$

*T_w_* is the local wall temperature at the microtube’s inner wall, and *T_f_* for the local bulk temperature at the tube’s axis. Additional average local *HTC* over the microtube length information is provided by:(21)$${HTC}_{ave}=\frac{1}{L}\int_{0}^{L}h_{l}dl$$

## 5. Results and Discussion

Table 4 lists the simulation conditions. The numerically simulated cases are solved for a time period of 1 s.

### 5.1. Validation

A separate validation of heat transfer, pressure drop, and evaporative flux is carried out because the two-phase flow in a microtube is a complicated process involving the heat transfer, pressure drop, and evaporation rate. Generalized correlations for 0.2 mm hydraulic diameter micro channels employing water fluid were proposed by Lee and Garimella [42]. Figure 6a depicts the results of a steady-state single-phase simulation with a chosen mesh at a heat flux of 50 W/cm^2^ and water entering the microchannel at a temperature of 300 K. The findings are in good agreement with the correlation. The second validation for the Nusselt number is conducted utilizing data from Tiwari and Mohrana [29]. A separate single flow with water fluid is investigated in a hydraulic tube of 0.6 mm at a heat flux of 45 KW/m^2^; the results are depicted in Figure 7a, which are consistent with the available literature data [29].

To build a correlation for heat transferal, a collection of 3899 data points covering 12 different wetting and non-wetting fluids with hydraulic diameter ranges of 0.16 to 2.92 mm, mass fluxes of 20 to 3000 Kg/m^2^ s, and heat fluxes of 0.4 to 115 W/cm^2^ was used by Bertsch et al. [43]. A separate steady-state simulation with n-heptane fluid at a mass flux of 140 Kg/m^2^ s with varying heat fluxes of 30–75 KW/m^2^ was calculated and compared with the HTC calculated by Bertsch correlation. The results are shown in Figure 7b and match with the literature [43].

An unsteady simulation was carried out with R141B at a fuel flow rate of 15 L/h and heat flux of 24 KW/m^2^, similar to the experimental condition of Yang et al. [44], and the results are shown in Figure 6c. The properties of R141B were taken from REFPROP [45]. The results show that till 0.04 s, the oscillation of the pressure drop with the present model and published data is similar; however, at 0.05 s and 0.11 s, the pressure drop peaks are opposite. This difference is due to two reasons. First, the fluid properties used by Yang et al. [44] may vary with temperature, and in the present research, the fluid properties are taken as constant. The second reason is due to mesh size. The liquid patches inside the vapor region are almost similar in qualitative comparison, as shown in Figure 7c.

**Figure 7 micromachines-14-01760-f007:**
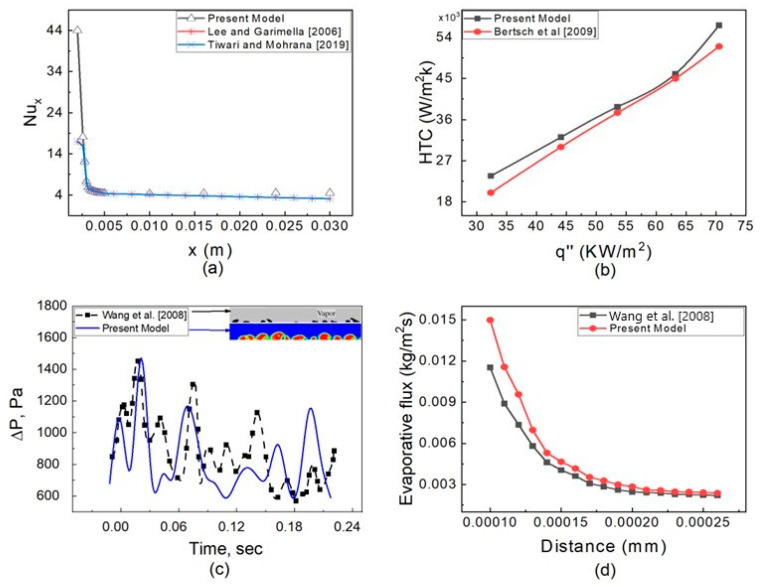
Validation of the present numerical model. (**a**) Validation of Nusselt number, Lee and Garimella [42], Tiwari and Mohrana [29]. (**b**) Validation of HTC, Bertsch et al. [43]. (**c**) Validation of pressure drop, Wang et al. [46]. (**d**) Validation of evaporative flux, Wang et al. [46].

Wang et al. [46] suggested a generalized model to calculate the evaporative flux at the interface with methanol fluid. Methanol’s characteristics were acquired from REFPROP [45]. Using the present model, we calculate the evaporative flux in a 0.6 mm ID open microtube filled with methanol. The results are shown in Figure 7d. The evaporative flux at the corner is high with the present model and this is may be due to mesh size across the radial direction, whereas the evaporative flux at the middle of the meniscus is similar in both the present model and the previous literature [46].

### 5.2. Mass Flow Rate Effect on Two-Phase Flow and Evaporation Rate 

At low fuel flow, the stable flow with dynamic meniscus inside the microtube is obtained, whereas a circular blue flame is anchored at the tube exit, as shown in Figure 2a. This blue flame indicates complete combustion due to the complete evaporation of liquid fuel at the interface [17,23]. The dynamic behavior of the evaporating meniscus is due to surface tension, vapor recoil, thermos-capillarity, viscous force, geometric constraints, and associated thin-film geometry [47]. At a high FFR, the nucleate bubble downstream of the evaporating meniscus was obtained. Similarly, at a low fuel flow rate, liquid droplets were not obtained; however, at a high fuel flow rate, liquid droplets with a random diameter were observed.

Figure 8a,b illustrates distinctive interface experimental and simulation results, respectively, with different time t_0_. This time is different due to the involvement of several factors in experimentation and simulation. However, the sub-time step is the same. The liquid film is observed in the simulated results, whereas in the experiments, it was not observed. Figure 8c shows the dynamics of evaporating meniscus in both experimental and simulated conditions with a maximum difference of 2%. The courant number for a low fuel flow rate was kept under 2.

At FFR > 30 µL/min, an unstable flow with the nucleate bubble in the liquid region and liquid droplets at the vapor region were formed. Figure 9 shows the growth cycle of the nucleating bubbles in both the experimental and simulation work. Figure 9a indicates that the nucleating bubble first drops from the wall, increases in size, sticks to the wall again, and finally smears the interface with an asymmetric geometrical configuration. However, Figure 9b indicates the nucleating bubble at the axis due to the axisymmetric solver used in the calculation. Figure 9c shows a good comparison of the nucleating bubble diameter observed in the experimental and simulated results against the normalized time. Figure 9d shows the detachment distance between the nucleating bubble and the liquid–vapor interface. The detachment distance found in the simulated results was not in accordance with the experimental results, which may be due to the use of an axisymmetric solver; however, the profile of the detachment distance is nonlinear, as in the experimental results. The courant number for a high fuel flow rate was kept under 0.1.

Figure 10a shows the two-phase flow configuration with a nucleating bubble in the downstream and a liquid droplet in the upstream both in an experimental and simulated analysis at an instantaneous time. Figure 10b shows the liquid droplet diameter at different FFRs with an error of 10%, which is acceptable for the simulation method.

Figure 11 shows the pressure drop across the interface, evaporation rate at the interface, and interface distance with flow time. The distinctive interface moves 0.1 mm in the microtube due to the vapor pressure difference across the interface, and this movement has a significant effect on the pressure drop and evaporation rate. This oscillatory behavior of the evaporating meniscus is due to many factors like vapor recoil, surface tension, and thermos-capillarity, as discussed by [47]. The pressure drop and evaporation rate have the same trend in flow time. If we look at Figure 7d, we can see that the maximum evaporative flux is found near the wall and this is due to high diffusion flux. The non-uniformity of diffusion flux at the meniscus results in strong evaporation near the wall.

### 5.3. Mass Flow Rate Effect on Heat Transfer

The impact of FFR and TB on HTC is discussed in this section. Figure 12 depicts the axial fluctuation of HTC in a microtube at 0.20, 0.60, and 0.90 s and Equation (20) is also used to calculate the average HTC. Figure 11 illustrates that the HTC increases with an increase in FFR for both subcooled and bulk boiling, and this relationship is also endorsed by Lie et al. [48]. However, Bertsch et al. [49] oppose the rise of HTC with an increased mass flow rate. Moreover, Basu et al. [50] revealed that HTC mainly depends on surface tension. At FFR ≤ 10 µL/min, i.e., for the condition of FL5TB5 and FL10TB10, there is a single peak observed at all time steps, as shown in Figure 12a–c, which is possibly due to a very small temperature gradient across the interface. The HTC for FL10TB10 is higher from FL5TB5 because higher temperature gradients cause the temperature to increase more quickly, which induces stronger evaporation and results in the movement of the interface to the outlet of the tube.

At increased FFR ≥ 30 µL/min, there are multiple peaks of HTC, as shown in Figure 12. The precipitous peak of HTC value 16,000 W/m^2^-K is found for FL50TB50 at 0.20 s, as shown in Figure 12a, while the remaining peaks are of the order of 12,000 W/m^2^-K and this is due to nucleating bubbles, as seen in Figure 9 and Figure 10. The same rapid peak is also revealed by Huang et al. [51]. The HTC for FL70TB70 first increases and then suddenly decreases due to two reasons. The first is due to the existence of a vapor bubble before the interface, which results in low heat transfer due to the low thermal conductivity of the quartz tube. The second is due to the presence of nucleating bubbles with low specific heat. The average HTC coefficient for five simulated settings during a time period of one second is shown in Figure 12d. The average HTC for FL5TB5 and FL10TB10 is less than 4000 W/m^2^-K, whereas it ranges from 5000 W/m^2^-K to 16,000 W/m^2^-K for FL50TB50 and FL70TB70. The nucleating bubble is the reason why the average HTC for FL30TB30 is between 2950 and 6550 with only two peak values of 12,500 W/m^2^-K and 14,500 W/m^2^-K at 0.64 s and 0.89 s, respectively.

In Figure 12, the combined effect of FFR and TB is presented, which makes it difficult to recognize the dominating factor on HTC peaks. We further segregated the simulation into two types: the first type has the same FFR and the TB was changed, whereas in the second type, the TB was kept constant and FFR was changed. Figure 13 shows the axial HTC with the same FFR and different TB at 0.20, 0.60, and 0.90 s. The single peak is observed for both FL5TB5 and FL5TB10 at all time steps. Multiple peaks of HTC are formed for FL5TB30 at 0.90 s; similarly, multiple peaks with a maximum value of 7500 W/m^2^-K and 7000 W/m^2^-K are found for FL5TB50 and FL5TB70, respectively. The HTC in the subcooled regions remains the same; however, at 0.60 s, the HTC is higher, which is due to the nucleate bubble in the liquid region. Figure 12 demonstrates that as the interface location approaches the tube’s exit, the values of HTC increase and this is due to an increase in the thermal gradient, which induces the nucleating bubble phenomenon that creates the liquid film, which forms high heat flux during liquid–solid thermal interaction. Figure 12d shows the average HTC at different intervals of time. The maximum average HTC is found for FL5TB70 at 0.16 s, whereas the other HTCs for the different conditions are superimposed at all intervals of time.

Figure 14 shows the axial HTC with different FFR and the same TB. The single peaks for both FL5TB5 and FL10TB5 are shown and multiple peaks for higher FFR are also shown. At 0.20 s, the maximum peak for 50 and 70 µL/min is 8000 and 11,200 W/m^2^-K, respectively, which shows that an increase in FFR increases the HTC. At 0.60 s, the maximum peak for 50 and 70 µL/min is 11,400 and 11,600 W/m^2^-K, respectively; however, this behavior of HTC is contradicted at 0.90 with 50 µL/min having a maximum HTC value of 11,500 W/m^2^-K, while at 70 µL/min, the maximum value of HTC is 8000 W/m^2^-K. Due to the existence of liquid droplets in the vapor area and nucleate bubbles in the liquid zone, there is a direct and inverse link between FFR and maximal HTC. These bubbles and droplets form the liquid film in the respective region, which increases and deteriorates the HTC, respectively. Figure 14d shows the average HTC at different intervals of time. After this time step, the average HTC for 50 µL/min is higher than 70 µL/min, which may be caused by flow phenomena inside the microtube. The average HTC and FFR have a direct relationship up to 0.6 s. The average value of HTC for 30 µL/min is greater than 50 µL/min and 70 µL/min at 0.60 s.

The discussion in this section leads to the conclusion that conjugate heat transfer and annular flow are the dominant processes in the microtube. This type of flow is also reported by Yen et al. [52]; however, they did not find a peak in the HTC calculation while investigating the convective boiling in a 0.51 mm ID tube with FC72 fluid. Furthermore, it is concluded that TB controls the peak position, whereas FFR controls the peak value of HTC.

### 5.4. Mass Flow Rate Effect on Pressure Oscillation and Volume Fraction

The influence of FFR is investigated on pressure oscillations and volume fraction. The results of three flow conditions, i.e., FL5TB5, FL30TB30, and FL70TB70, will be discussed. Figure 15 represents the pressure oscillation and spatial discretization of VF for FL5TB5. The pressure oscillations in the microtube are due to the rise in the vapor quality. Figure 14 shows that the increase in vapor quality decreases the two-phase flow density. This decrease in density further increases the velocity, and as a result, the pressure of the vapor phase increases. Wang et al. [53] estimated the capillary pressure in the stable flow using the following formula:(22)$$P_{c}=P_{v}-P_{l}$$
where *P_l_*, *P_c_*, and *P_v_* are liquid pressure, capillary pressure, and vapor pressure, respectively. The interfacial tension present at the contact causes capillary pressure. The instability in the molecules’ molecular force of attraction actually causes the interfacial tension. Figure 14 demonstrates that the capillary pressure varies over the course of the time step, which may be caused by conjugate heat transfer.

The pressure oscillation and VF for FL30TB30 are shown in Figure 16. In the microtube, nucleate boiling with an evicted liquid droplet from the interface occurs at high FFR. Figure 16a shows that liquid droplets escape from the interface at an axial location of 3.4 mm and have a pressure of 104.5 kPa, which is due to the high kinetic energy of the droplets. Similar to this, the nucleate bubble that occupies two-thirds of the microtube’s diameter has an extreme pressure of 101.8 kPa prior to a larger size and breaking down the liquid–vapor interface. Figure 16b shows a nucleation site with a vapor VF of 0.4 that has a pressure of 103.2 Kpa, while two small nucleating bubbles have a maximum pressure of 102.8 kPa. A single liquid droplet is seen at both 0.2 and 0.4 s, with an approximate pressure of 104.2 kPa. Figure 16c shows a very tiny micro droplets with a pressure of 105 kPa due to a small volume.

The pressure oscillation and VF for FL70TB70 are shown in Figure 17. The liquid droplet intensity breakout from the interface is higher than FL30TB30. The liquid slug is visible at 0.4 s and a train of the liquid droplet can be termed as Taylor bubbles. It is important to note that at 0.2 s, the nucleating bubble continues to grow and the liquid regime’s axial pressure continues to decline, whereas at 0.4 s, the nucleating bubble breaks up and the liquid regime’s axial pressure tends to increase. The rise in wall temperature may be the cause of the pressure drop, whilst the energy development in the nascent nucleating bubble is the cause of the increase. The increase in axial pressure can also be seen at 0.6 s, as shown in Figure 17c. A similar increase and decrease in axial pressure can also be seen in Figure 16a,d for FL30TB30; however, this increase or decrease is not as steep as in Figure 17a,c, and this may be due to an increase in the fuel flow rate for the latter case.

### 5.5. Fuel Flow Rate Effect on Ressure Drop

To determine the impact of fuel flow rate and wall temperature on the fluctuation frequency and amplitude, the data on pressure drop are analyzed. For each of the 13 cases listed in Table 4, the pressure drop data are presented in Figure 18a,c,e against a timescale of one second. Figure 18a demonstrates that the maximum pressure drop increases from 500 Pa to 2800 Pa when the fuel flow rate increases from 5 µL/min to 70 µL/min. This increase in extreme pressure drop is caused by greater flow resistance at an upper mass flow rate. Wang et al. [54] investigated the pressure drop fluctuation of two-phase flow in a 0.5 mm microtube and reported similar results. However, Yen et al. [52] disagreed with this assertion and came to the conclusion that the pressure drop features in microtubes remain unchanged. Figure 18c depicts that the maximum pressure drop oscillations are up to 900 Pa if the FFR is kept constant, whereas Figure 18e demonstrates that the maximum pressure drop oscillations are up to 2600 Pa if the FFR is changed and TB is kept constant. This depicts that fuel flow rate has a dominant impact on pressure decrease.

The results of the pressure drop in this part represent the entire pressure drop during the two-phase flow. As a result, the pressure drop entails three components: pressure drop owing to acceleration, which is also influenced by void fraction and density changes, pressure drop due to wall friction, and pressure drop caused by abrupt contraction and expansion in the microchannel [55]. According to Kawahara et al. [55] the frictional pressure drop accounts for the majority of the total pressure loss. The pressure reduction in the current study is mostly caused by phase shift, the presence of the nucleating bubble, and the liquid droplet.

The frequency spectrum of the pressure drop was obtained for all 13 cases as described in Table 4 and shown in Figure 18b,d,e. Figure 18b shows the FFT analysis for flow conditions with different flow rates and different temperature gradients. For 5 µL/min, the high amplitude of 380 is found at a low frequency of 2 Hz while at a high frequency, the amplitude is 50. For 70 µL/min, the amplitude was 620 at a low frequency, while at a higher frequency, the maximum amplitude is 300. Figure 18d shows the FFT analysis for 5 µL/min with different temperature gradients. As seen in Figure 18d, at low frequency, the amplitude is 400 for all temperature gradients, while at higher frequency, the amplitude is around 50 for all temperature gradients. This concludes that the high amplitude is due to the mass flow rate. Bogojevic et al [56] and Wang et al. [54] captured the high-frequency low-altitude fluctuation as well as the low-frequency high-amplitude fluctuation. While the high-frequency, low-amplitude fluctuations were mostly caused by bubbles and liquid droplets, the low-frequency, high-amplitude fluctuations were thought to be caused by two-phase vapor–liquid separation. This is the reason that at 5 µL/min, the amplitude fluctuation is not observed due to the absence of liquid droplets.

The pressure drop’s coefficient of variation (COV) is also used to study pressure drop oscillations. This coefficient is used to determine variability and diversity of the data from the average value. It is the ratio of the mean value ΔP_avg_ to the standard deviation ΔP_rms_. Singh et al. [57] previously employed this analysis technique. The coefficient of variation is plotted in Figure 19. The COV increases with the rise in the fuel flow rate for the first group till 50 µL/min, and then decreases, and a second-order polynomial curve is obtained. For the third group, the COV is almost similar at all flow rates and a linear fitted curve was obtained.

## 6. Conclusions

The numerical model analyses and thoroughly discusses the flow boiling of liquid n-heptane in a microtube at various fuel flow rates. In the experimental work, nucleating bubbles, liquid bubbles, and two-phase flow patterns are all observed. To examine heat transfer, pressure oscillation, pressure drop, and evaporative flux at the meniscus, numerical simulations have been carried out at a fuel flow rate comparable to that employed in the experimental investigations. The main outcomes of this study are as follows:The steady flow with a dynamic meniscus was obtained at FFR < 10 µL/min and the highest wall temperature was 1050 K. In the microtube, an unstable flow with nucleate bubbles and liquid droplets predominates at an FFR of 30 to 70 µL/min.It is found that the pressure drop across the meniscus and the interface distance of the meniscus from the exit of the microtube affect the evaporative flux at the meniscus.One peak of HTC was obtained when the FFR was less than 10 µL/min, but at high fuel flow rates of 30 to 70 µL/min, numerous peaks of HTC were obtained at various time steps. The location of the peak was determined based on the temperature boundary condition, whereas the peak of HTC is dependent on the fuel flow rate.At a low fuel flow rate of 10 µL/min, the capillary force was found to be dynamic. The pressure oscillation was found at 30–70 µL/min. In the liquid regime, the axial pressure tends to rise during nucleating bubble breakup and decrease during bubble growth.At various fuel flow rates, the oscillation of the pressure drop was also obtained in the microtube. At all flow rates, both high-frequency low-amplitude and low-frequency high-amplitude pressure fluctuations were observed. The liquid n-heptane fuel flow rate is mostly responsible for the large amplitude.The fuel flow rate remained constant while the temperature boundary changed, or it remained constant while the fuel flow rate changed, although the slope of the fitted curve was larger in the latter instance.

## Figures and Tables

**Figure 1 micromachines-14-01760-f001:**
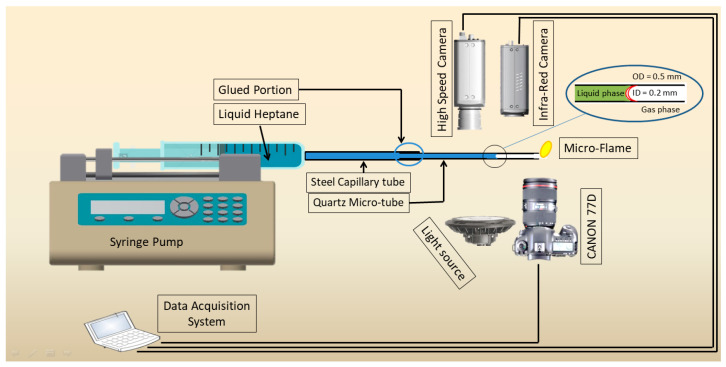
Sketch of the experimental setup.

**Figure 2 micromachines-14-01760-f002:**
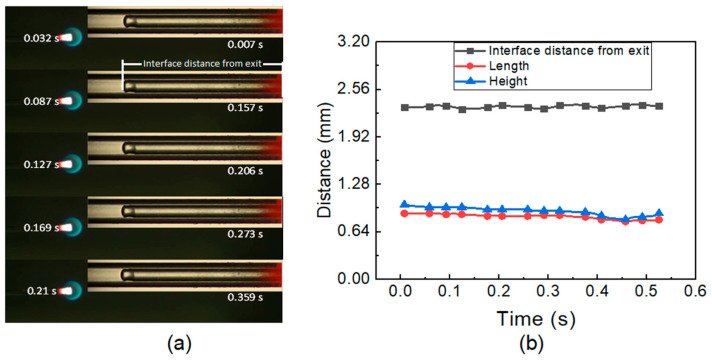
(**a**) Flame picture, flow pattern at 5 micro. (**b**) Interface distance and flame characteristics at low FFR.

**Figure 3 micromachines-14-01760-f003:**
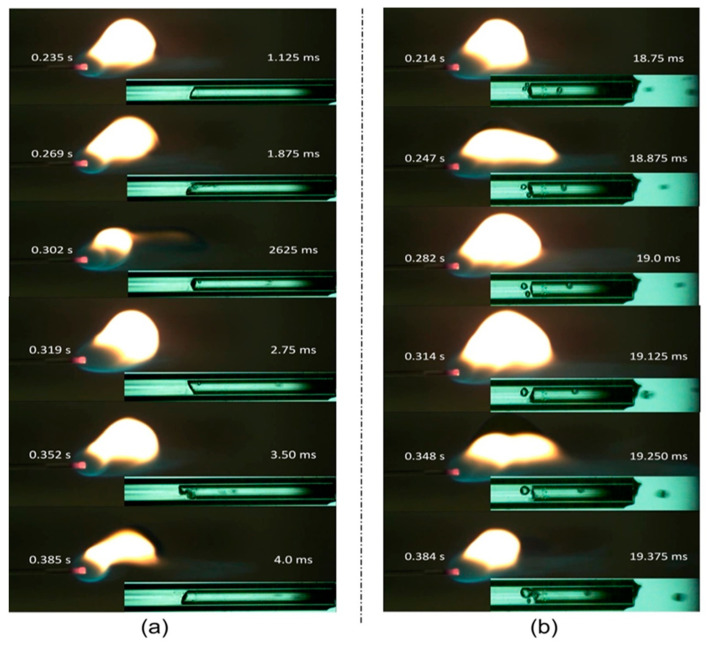
Flame characteristics and evaporation phenomena: (**a**) 30 µL/min, (**b**) 70 µL/min.

**Figure 4 micromachines-14-01760-f004:**
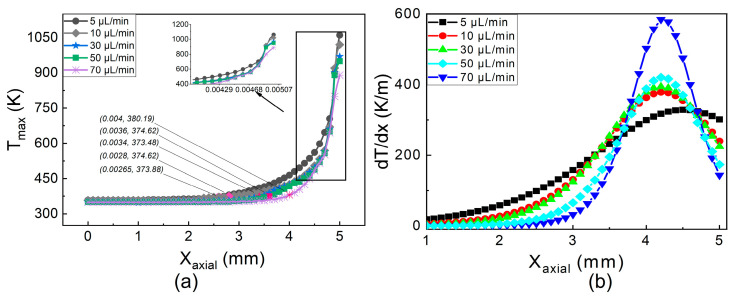
Temperature distribution. (**a**) Temperature profile. (**b**) Thermal gradient.

**Figure 5 micromachines-14-01760-f005:**
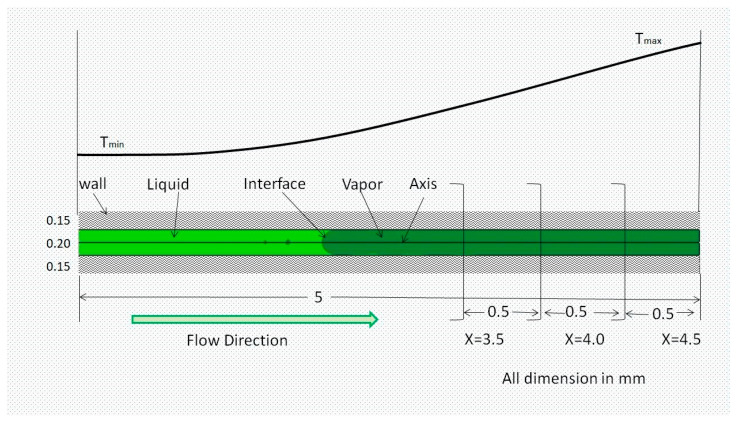
Schematic of the capillary tube.

**Figure 6 micromachines-14-01760-f006:**
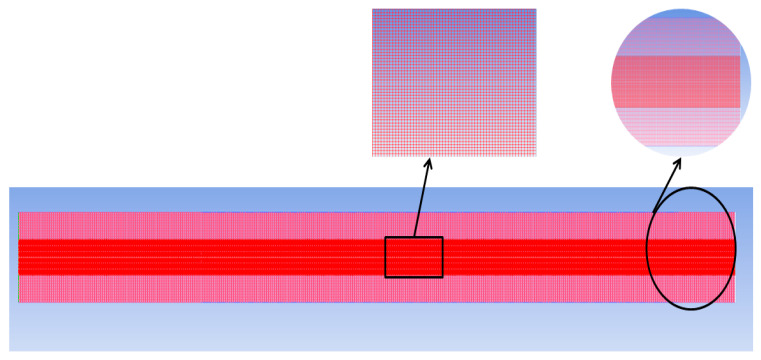
Computational grid: liquid and solid region.

**Figure 8 micromachines-14-01760-f008:**
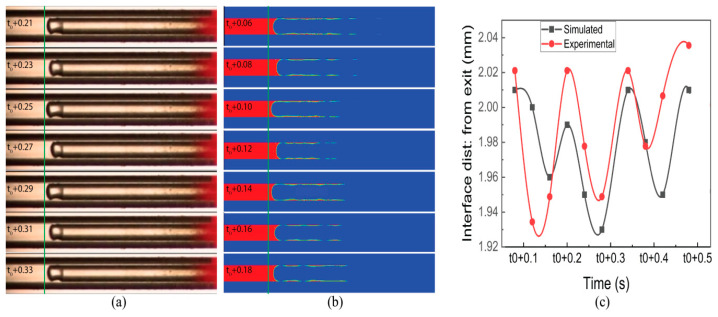
(**a**) Experimental results of interface dynamics. (**b**) Simulated results of interface dynamics. (**c**) Comparison of interface distance in experimental and simulated results (time in sec).

**Figure 9 micromachines-14-01760-f009:**
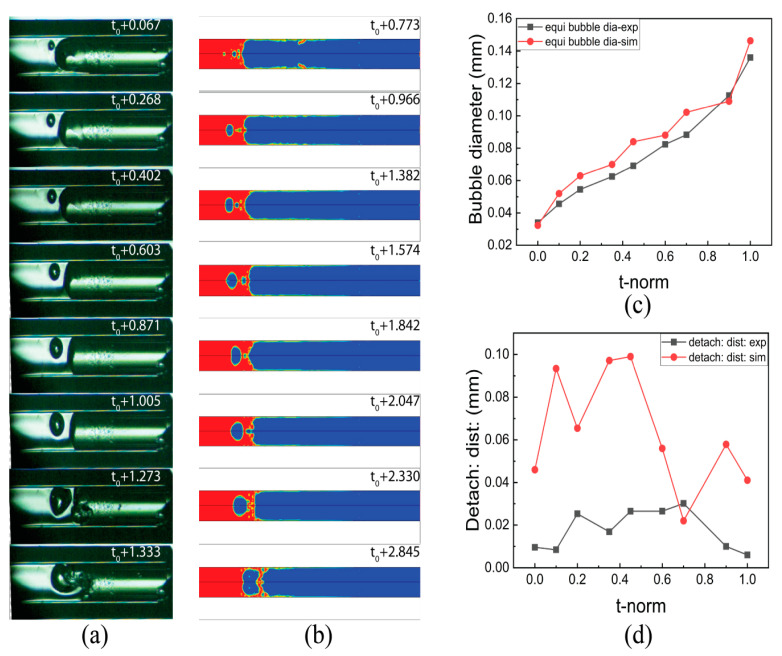
Nucleating bubble: (**a**) experimental result, (**b**) simulation result, (**c**) equivalent bubble diameter vs. normalize time, and (**d**) detachment distance vs. normalize time (time in m/s).

**Figure 10 micromachines-14-01760-f010:**
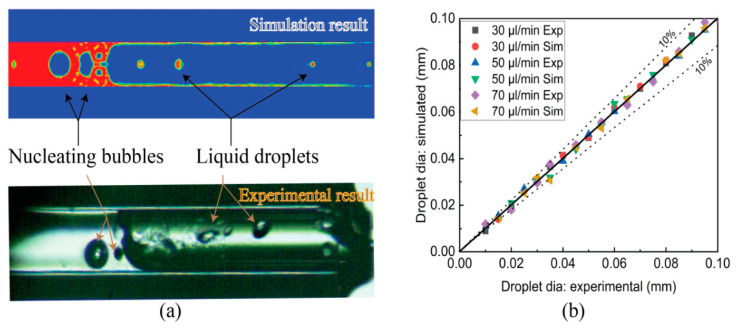
(**a**) Two-phase flow with nucleate bubble and liquid droplet at an instantaneous time. (**b**) Droplet diameter experimental vs. simulation at different fuel flow rates.

**Figure 11 micromachines-14-01760-f011:**
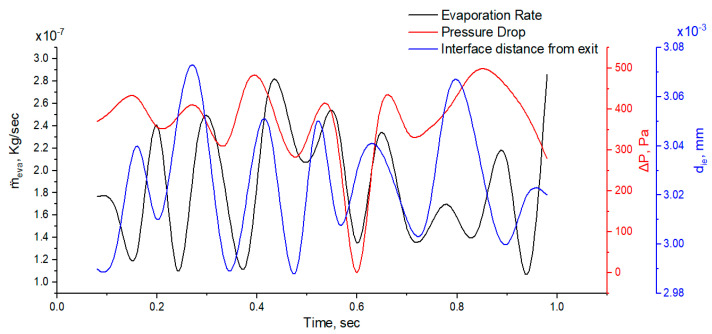
Plot of interfacial distance, pressure drop, and evaporation rate vs. time.

**Figure 12 micromachines-14-01760-f012:**
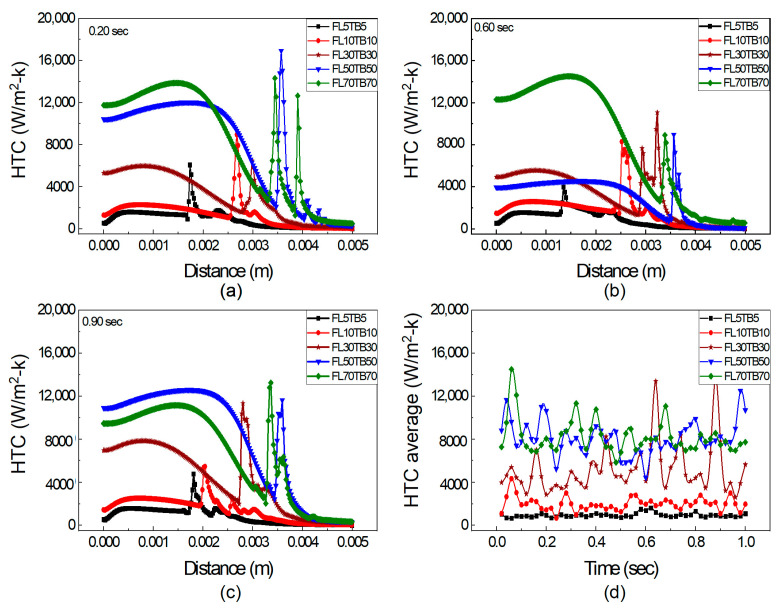
HTC at axial distance with different FFR and TB: (**a**) 0.20 s, (**b**) 0.60 s, and (**c**) 0.90 s. (**d**) Average HTC vs. flow time.

**Figure 13 micromachines-14-01760-f013:**
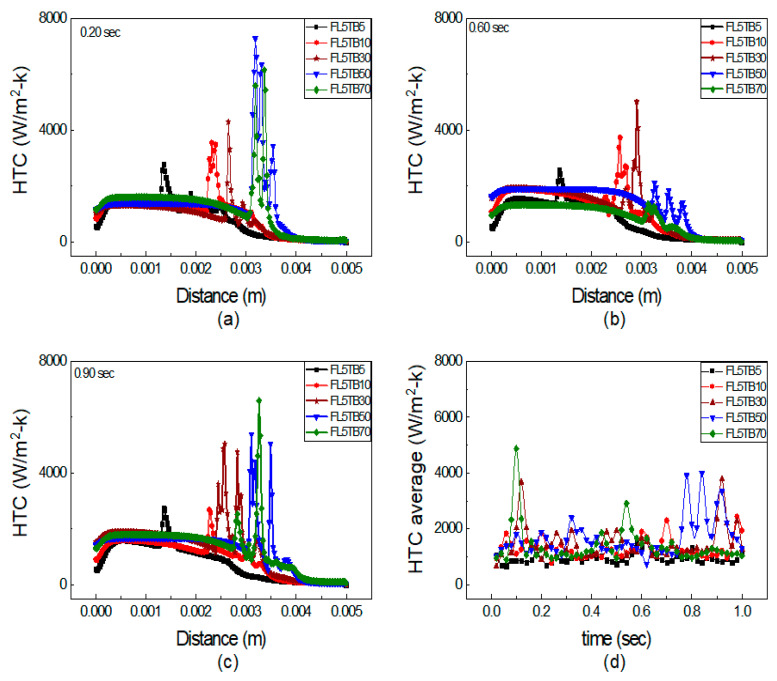
Axial HTC for same FFR and different TB: (**a**) 0.20 s, (**b**) 0.60 s, and (**c**) 0.90 s. (**d**) Average HTC vs. flow time.

**Figure 14 micromachines-14-01760-f014:**
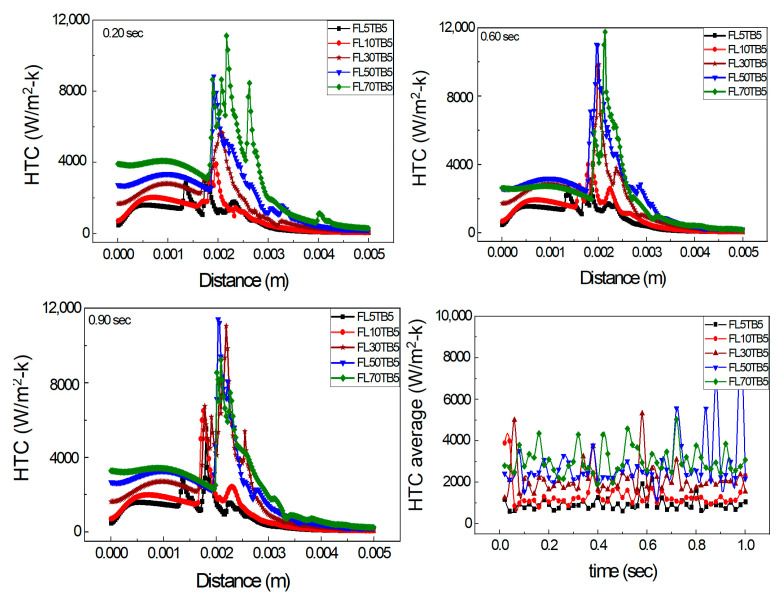
Axial HTC for different FFR and same TB: (**a**) 0.20 s, (**b**) 0.60 s, and (**c**) 0.90 s. (**d**) Average HTC vs. flow time.

**Figure 15 micromachines-14-01760-f015:**
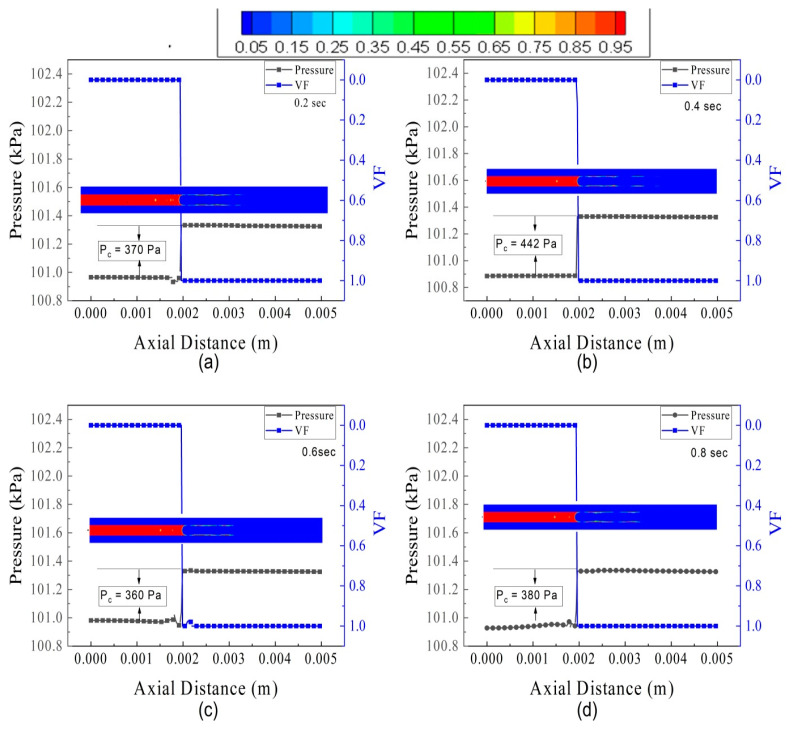
Pressure and volume fraction in the microtube at different intervals of time for FL5TB5: (**a**) 0.20 s, (**b**) 0.40 s, (**c**) 0.60 s, and (**d**) 0.8 s.

**Figure 16 micromachines-14-01760-f016:**
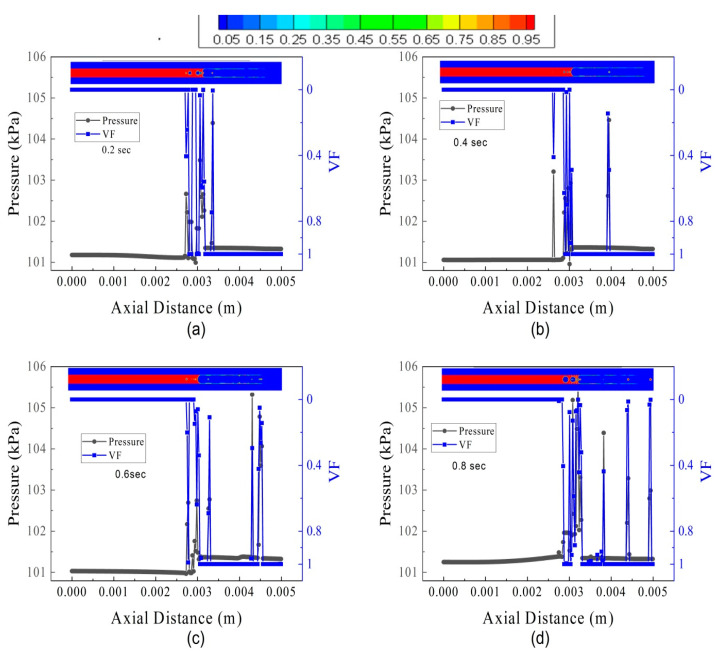
Pressure and volume fraction in the microtube at different intervals of time for FL30TB30: (**a**) 0.20 s, (**b**) 0.40 s, (**c**) 0.60 s, and (**d**) 0.8 s.

**Figure 17 micromachines-14-01760-f017:**
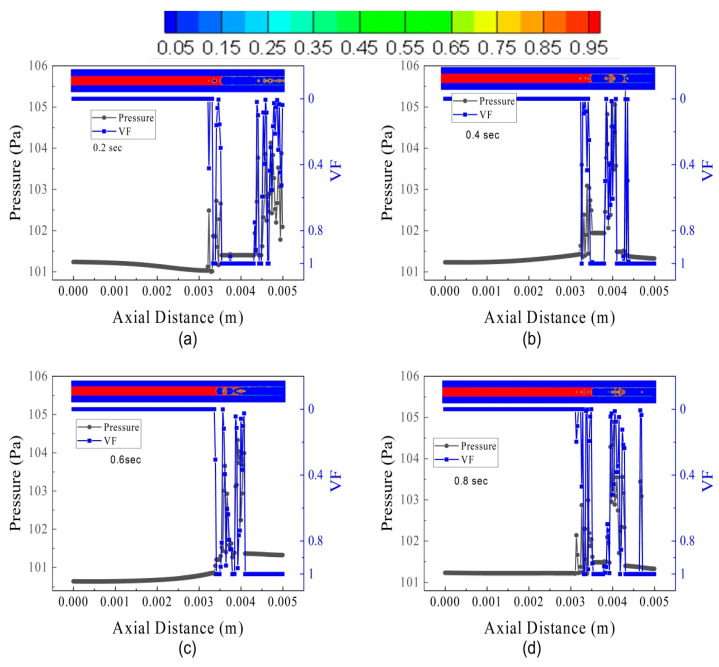
Pressure and volume fraction in the microtube at several intervals of time for FL70TB70: (**a**) 0.20 s, (**b**) 0.40 s, (**c**) 0.60 s, and (**d**) 0.8 s.

**Figure 18 micromachines-14-01760-f018:**
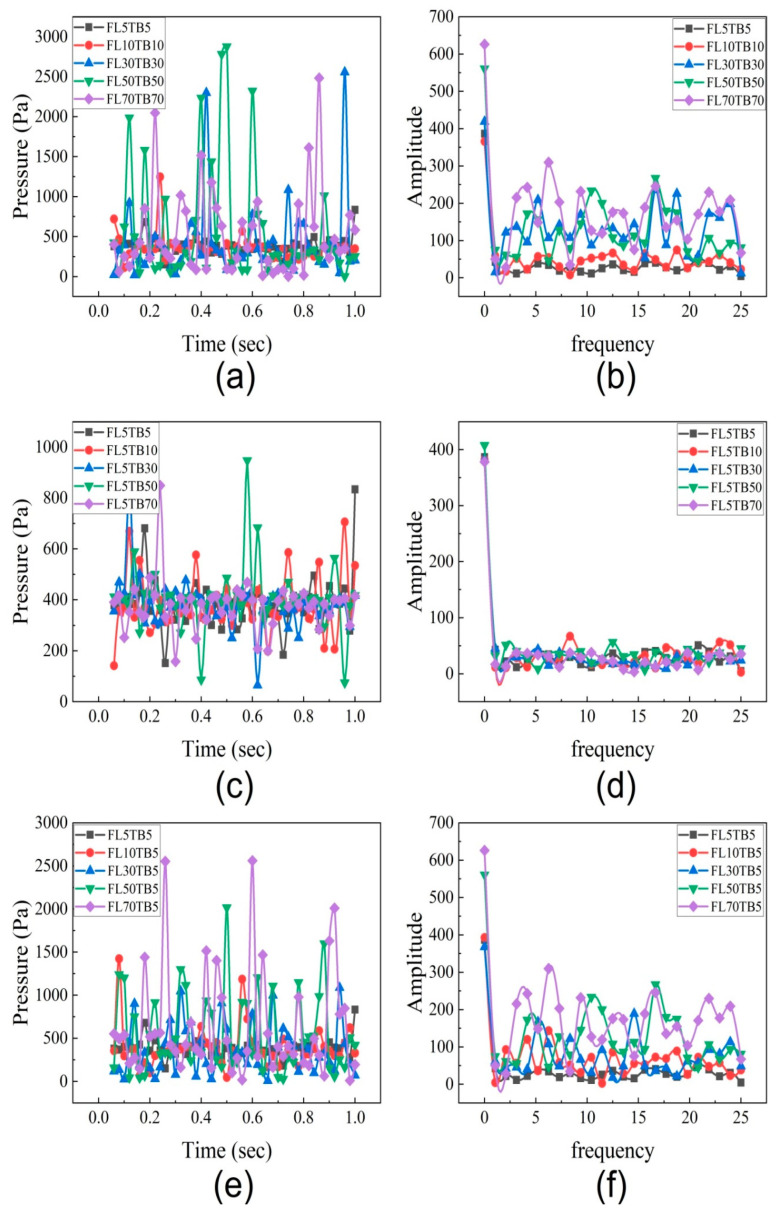
Pressure drop data for different flow conditions. (**a**) Pressure drop with different flow rate and different temperature boundary. (**b**) Pressure drop frequency spectrum with different flow rate and different temperature boundary. (**c**) Pressure drop with same flow rate and different temperature boundary. (**d**) Pressure drop frequency spectrum with same flow rate and different temperature boundary. (**e**) Pressure drop with different flow rate and same temperature boundary. (**f**) Pressure drop frequency spectrum with different flow rate and same temperature boundary.

**Figure 19 micromachines-14-01760-f019:**
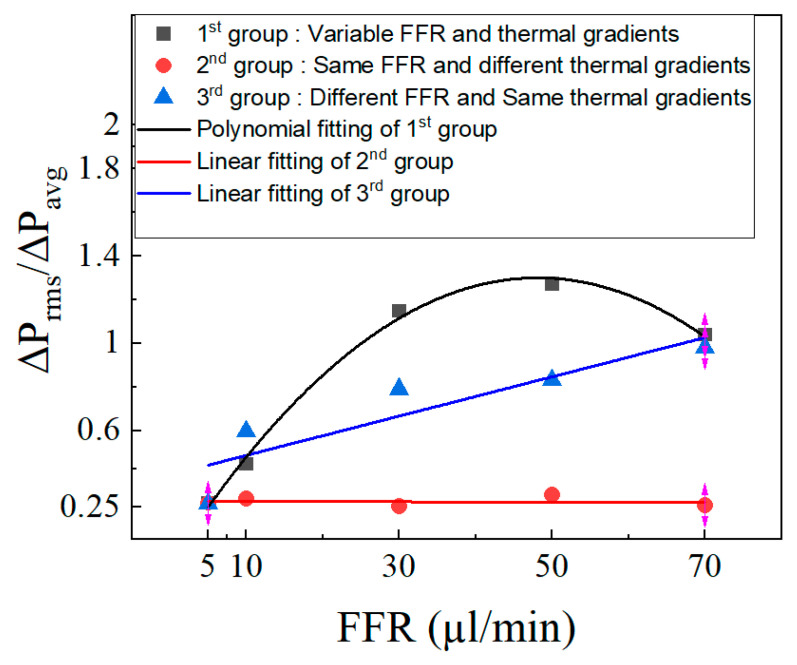
Coefficient of variation at different FFR.

**Table 1 micromachines-14-01760-t001:** Heptane properties [35].

Property	Value
Density (kg/m^3^)	666.95 at 295.45 K
Thermal conductivity (W/m-K)	0.12746
Thermal capacity (J/kg)	2298.4
Viscosity (kg/m-s)	0.000331
Vapor molecular weight	100
Evaporation latent heat (J/kg)	3.506 × 10^5^
Saturated vapor pressure (Pa)	5316 at 295.45 K
Accommodation coefficient	1

**Table 2 micromachines-14-01760-t002:** Mesh dependency result.

Grid No.	Grid Size	Average HTC (W/m^2^-K)	e%	Evaporative Flux at Meniscus (Kg/m^2^-s)	e%	ΔP (Pa)	e%
1	64,500	967	-----	8.63 × 10^−9^	----	1324	----
2	34,000	960	0.72	8.56 × 10^−9^	0.81	1315	0.67
3	22,000	945	2.2	8.5 × 10^−9^	1.51	1290	2.5

**Table 3 micromachines-14-01760-t003:** Volumetric flow rate of fuel and corresponding mass flow rate.

FFR (µL/min)	5	10	30	50	70
MFR (mg/s)	0.056	0.113	0.343	0.58	0.797

**Table 4 micromachines-14-01760-t004:** Flow conditions.

Case. No	Case Name	Investigation	Remarks
1	FL5TB5	Experiment and Simulation	Boundary condition retrieved from experiment with 5 µL/min of mass flow rate
2	FL10TB10	Experiment and Simulation	Boundary condition retrieved from experiment with 10 µL/min of mass flow rate
3	FL30TB30	Experiment and Simulation	Boundary condition retrieved from experiment with 30 µL/min of mass flow rate
4	FL50TB50	Experiment and Simulation	Boundary condition retrieved from experiment with 50 µL/min of mass flow rate
5	FL70TB70	Experiment and Simulation	Boundary condition retrieved from experiment with 70 µL/min of mass flow rate
6	FL10TB5	Simulation	Flow rate is 10 µL/min while boundary condition is from case 1
7	FL30TB5	Simulation	Flow rate is 30 µL/min while boundary condition is from case 1
8	FL50TB5	Simulation	Flow rate is 50 µL/min while boundary condition is from case 1
9	FL70TB5	Simulation	Flow rate is 70 µL/min while boundary condition is from case 1
10	FL5TB10	Simulation	Flow rate is 5 µL/min while boundary condition is from case 2
11	FL5TB30	Simulation	Flow rate is 5 µL/min while boundary condition is from case 3
12	FL5TB50	Simulation	Flow rate is 5 µL/min while boundary condition is from case 4
13	FL5TB70	Simulation	Flow rate is 5 µL/min while boundary condition is from case 5

## Data Availability

No data available.

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
