# Peer review of "Flow Boiling of Liquid n-Heptane in Microtube with Various Fuel Flow Rate: Experimental and Numerical Study"

_micromachines, 2023, doi:10.3390/mi14091760_

Round 1
Reviewer 1 Report
The submitted work details experimental and numerical analysis on n-heptane flow boiling in a microtube. I believe the present manuscript can be considered for publication after addressing the following issues.
1. The title needs to be revised to show the research focus of this paper: flow boiling;
2. The format of the figures should be consistent.
3. Writing needs to be improved.
Needs to be improved.
Author Response
Authors Responses to Reviewers Comments
The reviewers' comments were insightful. We made every effort to update the manuscript in accordance with the suggested modifications, which significantly improved our paper. We have modified and rewritten the paper based on the reviewers' suggestions and comments. In the revised version, the added content is marked with underlined blue colored.
Reviewer #1:
The submitted work details experimental and numerical analysis on n-heptane flow boiling in a microtube. I believe the present manuscript can be considered for publication after addressing the following issues.
⦁ The title needs to be revised to show the research focus of this paper: flow boiling;
Response: The authors are thankful to the reviewer for highlighting this important aspect. The title of the paper has been changed in the revised version of the manuscript. The new title is
“Flow boiling of liquid n-heptane in microtube with various fuel flow rate; Experimental and Numerical study”
⦁ The format of the figures should be consistent.
Response: The authors are thankful to the reviewer for this important suggestion. The necessary figures have been modified in the revised version of the manuscript.
⦁ Writing needs to be improved.
Response: Thank you for the incisive review. The writing has been improved and corrected in the revised manuscript.
Reviewer 2 Report
In this paper, the authors have investigated the evaporation of liquid hydrocarbon n-heptane numerically and experimentally. The investigations are interesting. Nevertheless, the authors have to give clarity on the following comments point by point.
Q1. The abstract must also have vital qualitative results in terms of %. viz. FFR vs pressure peaks
Q2. Figure 1 protray the capillary tube in a horizontal position, agreed as it is a microtube. How the red micro flame is cited in the figure? Is this red flame indicate a section in the figure?
Q3. In Figure -b the legend indicationg 5 micro, 10 micro is not complete. Correct this.
Q4. Figures a and b could be one below the other so that their width is bigger and for a better readability.
Q5. In figure 3b, at a distance say 0.0043 m the temperature gradient is peak for all the flow rates. Authors have to infer this phenominon in a better way in terms of % change and w.r.t FFR.
Q6. In figure 7a the staggered data at distance , 0.005 will be visible only when the figure dimensions are appropriate. Correct this
Q7. Correct the units in line 310, 407, 435 and also in the entire manuscript.
Q8. In lines 311 and 317, authors say that data matches with the literature. where are the references?
Q9. Even in the figures of 7 the references have to be mentioned.
Q10. Authors mention in line 335, "At a low FFR, the stable flow with dynamic meniscus was obtained", how do you justify this phenominon. Is there a reference to Reynold number to it?
Q11. Results and discussion is good, appreciated, however figures 12 to18 could be displayed better, for the proper visibility
Q12. Authors have to indicate the pressure peaks with respect to axial distance as well, i.e at multiple sections of the micro tube. This inference will attaract the readers.
Author Response
Dear Reviewer,
Thank you very much for your valuable feedback. The response file is attached.
Regards,

Reviewer 3 Report
The authors experimentally observed the flame stability and numerically studied the evaporation and boiling dynamics of n-heptane liquid in a capillary tube, under different flow rates. The results seem solid, but the writing and analyses are poor. I strongly recommend the authors to carefully revise it. The following issues should be addressed clearly.
1. For the figures, the legend should be clearly given, especially when subfigures exist.
2. What are the two time in Fig.1a, i.e., 0.032s and 0.007s etc.? What is the point of putting them together if time is different? What is interface height in Fig.1b?
3. Fig.3 jumps to Fig.5 without Fig.4, but some analyses are given based on Fig.4.
4. The title of section 5.1 is at the right side of table 4.
5. The pressure has big difference with the result from Wang et al., why is that?
6. The authors declare “Boundary condition retrieved from experiment” in Table 4, what is it exactly?
7. The experiment is 3D, but the simulation is axisymmetric 2D. Are they comparable?
8. The authors analyzed a lot about the pressure drop. How does the pressure drop dynamics influence the flame stability?
Should be improved, especially for the analyses.
Author Response

(The authors gave the same response as above.)

Round 2
Reviewer 3 Report
The reviewer appreciates the authors for their revision. However, the figure sequence is still messy after I commented it before, which certainly affects the analysis. Please be careful.